# Influence of the Addition of Recycled Aggregates and Polymer Fibers on the Properties of Pervious Concrete

**DOI:** 10.3390/ma16155222

**Published:** 2023-07-25

**Authors:** Oskar Mitrosz, Marzena Kurpińska, Mikołaj Miśkiewicz, Tadeusz Brzozowski, Hakim Salem Abdelgader

**Affiliations:** 1Faculty of Civil and Environmental Engineering, Gdańsk University of Technology, Narutowicza 11/12, 80-233 Gdańsk, Poland; oskar.mitrosz@pg.edu.pl (O.M.); mikolaj.miskiewicz@pg.edu.pl (M.M.); 2Keller Polska sp. z o.o., Poznańska 172, 05-850 Ożarów Mazowiecki, Poland; tadeusz.brzozowski@keller.com; 3Faculty of Engineering, Department of Civil Engineering, University of Tripoli, Tripoli 13932, Libya; h.abdelgader@uot.edu.ly; 4Libyan Authority for Scientific Research, Tripoli P.O. Box 80045, Libya

**Keywords:** pervious concrete, waste construction materials, recycled aggregate, polymer fibers, decarbonization in concrete production

## Abstract

The aim of the study was to check the possibility of reusing aggregate from recycled concrete waste and rubber granules from car tires as partial substitution of natural aggregate. The main objective was to investigate the effects of recycled waste aggregate modified with polymer fibers on the compressive and flexural strength, modulus of elasticity and permeability of pervious concrete. Fibers with a multifilament structure and length of 54 mm were deliberately used to strengthen the joints among grains (max size 31.5 mm). Eight batches of designed mixes were used in the production of pervious concrete at fixed water/binder ratio of 0.34 with cement content of 350 kg/m^3^. Results showed that the use of recycled concrete aggregate (8/31.5 mm) with replacement ratio of 50% (by weight of aggregate) improved the mechanical properties of pervious concrete in all analyzed cases. Whereas the replacement of 10% rubber waste aggregate (2/5 mm) by volume of aggregate reduced the compressive strength by a maximum of 11.4%. Addition of 2 kg/m^3^ of polymer fibers proved the strengthening effect of concrete structure, enhancing the compressive and tensile strengths by a maximum of 23.4% and 25.0%, respectively. The obtained test results demonstrate the possibility of using the recycled waste aggregates in decarbonization process of pervious concrete production, but further laboratory and field performance tests are needed.

## 1. Introduction

Pervious concrete (PC) is a form of concrete with a special structure and is made from strictly controlled amounts of water and cementitious materials (binders) used to create a paste that forms a coating around coarse aggregate. Unlike conventional concrete, the mixture has little or no fine aggregate, creating substantial void ratio. Using sufficient paste to coat and bind the aggregate particles together creates a system of highly permeable, open, and interconnected voids, allowing water and gases to permeate through its structure rapidly. Both the low binder content and high porosity reduce strength compared to conventional concrete, but sufficient strength is readily achieved for many applications. Therefore, PC is an effective stormwater management tool to reduce the volume of stormwater runoff and the concentration of pollutants [1]. Pervious concrete is also widely recognized as a sustainable building material used in pavement engineering for parking lots and road surfaces, and it has variety of the advantages, i.e., reducing noise, impact of the urban heat island effect, water accumulation, improving stormwater quality, and recharging groundwater supplies [2,3]. From past studies it was observed that pervious concrete mix proportions selected by an experimental basis vary from to region to region. However, the numbers of proposed studies on mix design of pervious concrete are very less [4,5,6,7].

The storage of massive quantities of waste rubber tires or construction materials coming from demolition activities, especially in urban areas, is one of the global largest sustainability challenges and environmental concerns [8,9,10,11,12]. Another issue is the depletion of natural resources for aggregates used in concrete manufacturing, thus even partial substitution of natural components in concrete is worth the effort. The reuse of waste materials in construction has benefits, not only by the reduction of waste generated that are generally disposed in landfills, but also in preserving natural resources, reducing the impacts associated with their extraction. Substituting the natural aggregate with recycled one can result in general in a considerable increment in permeability coefficient with adversely influencing mechanical properties [13,14]. However, regardless of the type of recycled concrete aggregate (i.e., waste crushed concrete) an almost linear relationship between the compressive strength and void ratio, and between permeability and void ratio is observed in pervious concretes [15] Results of the experiment showed that the use of recycled coarse aggregate significantly decreases the workability (slump) of concrete because of the recycled material’s high-water absorption and rough surface, but this can be neutralized by pre-wetting aggregate or modifying concrete with latex polymer [16].

Addition of recycled waste tire rubbers as fine aggregate in PC had a negative impact on mechanical properties [17], which is consistent with scientific studies [18,19] showing significant strengths decrease with a rubber-to-aggregate replacement. Incorporating rubber particles is detrimental to compressive and tensile strengths but may remain unaltered or even improved if cement composite is simultaneously treated with fibers [2,20,21,22,23,24,25].

Typical pervious concrete, as a mixture of uniform-sized coarse aggregates and cement, is combined at a relatively low water/binder ratio (w/b = 0.27 ÷ 0.34) [26]. The cement content should be just enough to coat the aggregate particles with a thin layer (below 200 micrometer). Excessive binder content may seal the voids between aggregate particles and significantly reduce permeability of the concrete [27]. The important mechanical properties of typical and modified pervious concrete mixes based on a recent review of the literature are shown in Table 1. In general, the unit weight of pervious concrete is approximately 1600–2000 kg/m^3^, which is already close to the upper limit of lightweight concrete. The air void content of the hardened PC can range from 15 to 35% allowing a drainage rate of 81 to 730 L/min∙m^2^ (permeability, k = 0.14 to 1.22 cm/s), with typical compressive strengths of 2.8 to 28 MPa [28]. As the permeability plays a crucial role in performance of pervious concrete, a high porosity is targeted. Due to the porous structure, the integrity and strength of PC depend mainly on the bonding between the coarse aggregate and the thin cover layer of the binder matrix, the-so-called interfacial transition zone (ITZ) [29]. Therefore, in general, the more porous and thinner the ITZ is, the lower the compressive strength. However, this may be opposed by replacing coarse with the smaller size aggregate. Adding a small amount of fine sand (approx. 7% by weight of aggregate) to the mix can significantly improve the concrete strength, but correspondingly will decrease the flow rate of water throughout the PC body [27]. Therefore, the best alternative solution to obtain higher compressive strength without adversely affecting the void ratio and permeability is to improve and densify the ITZ with mineral additives (i.e., nano-silica [30,31], silica fume [32], metakaolin [33]). Also, addition of latex polymer significantly improves workability and strength of pervious concrete while keeping its high porosity and permeability [16]. It is worth noticing that designing the mix proportions of PC with controlled permeability, workability and strength parameters is highly restricted to differences in components, variety of admixtures and additives, and influenced by the forming process.

The aim of the present study was to check the possibility of using aggregate from recycled concrete waste and rubber granules from car tires. The main objective was to develop and confirm with laboratory testing a ready-mix of pervious concrete, ensuring rational management of natural resources alongside with meeting the durability and strength dedicated requirements for further applications. To strengthen the concrete structure, special polymer fibers with a multifilament structure were used. In the research, fibers with a length of 54 mm were deliberately selected so that they could join a minimum of two aggregate grains (max size 31.5 mm). It was expected that such a solution would allow to modify and strengthen the structure without reducing the permeability of concrete. This is a new solution that requires further research.

Prior to the test program and experiment, initial mixes and previous research were analyzed to evaluate the feasible mixing ratios. Finally, the water-to-binder ratio was set at 0.34. The manuscript provides results of the first study to investigate the effects of reusing waste materials like recycled concrete and rubber waste aggregate on PC material properties. Possibilities of managing this type of waste are being sought so that it does not linger in landfills, but above all, the aim is to protect deposits of natural aggregates.

## 2. Materials and Methods

### 2.1. Materials

Materials used in the experiment (see Figure 1) included fine aggregates (0/2 mm), natural coarse aggregates (8/16 mm), recycled concrete aggregates (8/31.5 mm) and rubber waste aggregates (2/5 mm). Grain size composition (sieve analysis) is presented in Figure 2 and Table 2. Portland cement CEMI 42.5R was used with chemical and physical characteristics presented in Table 3.

To improve the overall performance of designed pervious concrete, a MC Bauchemie PowerFlow 2650 (MC-Bauchemie Müller GmbH & Co. KG, Bottrop, Germany) superplasticizer of 0.4% (by binder weight) was applied within the recommended dosage range. Due to the action of superplasticizer that causes high dispersion of cement grains, the workability of the mix was significantly improved, enabling efficient process from production to application. As an innovative solution keeping high-corrosion resistance, Polyex Mesh 2000 (EXPORPLÁS – Indústria de Exportação de Plásticos, S.A., Cortegaca OVR, Portugal), polymer fibers of 54 mm length (see Figure 1) were added to the designed mixtures to find an alternative for steel reinforcement. The fibers were characterized by the following parameters: tensile strength 650–750 MPa, Young’s modulus 5.90 GPa. The aim of the research was to determine the potential effects of recycled concrete aggregate with a replacement ratio of 50% (by weight of aggregate), rubber waste aggregate with a replacement ratio of 10% (by volume of aggregate) and fibers set at 2 kg/m^3^, influencing the mechanical properties of pervious concrete. The mix proportions for the laboratory-produced PC mixtures are presented in Table 4.

### 2.2. Preparation of Test Specimens

According to the mix proportions of PC listed in Table 4, the components were added to a mechanical mixer. Water, admixture, and cement were then evenly mixed to obtain a homogeneous grout, followed by fine aggregate, coarse aggregate (without pre-wetting), and polymer fibers. After thorough mixing, the PC specimens required for each test were cast with compaction by 20 blows with a tamper. No vibrating nor rodding were used. Along with each mix from J1 to J8, a certain number of specimens were prepared to determine compressive, tensile, and flexural strengths and modulus of elasticity of concrete (see Table 5 and Figure 3). The specimens were covered with construction film overnight for 24 h, demolded and subsequently placed in a standard curing room with a controlled temperature of 20 ± 2 °C and a RH ≥ 95%.

### 2.3. Test Methods and Data Analysis

According to code EN 12390-3 [46], a compressive axial load was applied to the cubic samples after 7, 28 and 56 days of curing at a continuous rate until failure occurred. The compressive strength was determined by dividing the maximum load by the cross-sectional area of the specimen (f_cm_). Tensile splitting strength was checked on cubic and cylindrical samples following EN 12390-6 [47] (Figure 4). A diametric compressive load was applied along the length of the sample at a continuous rate until failure occurred. This loading induced tensile stresses on the plane containing the applied load, causing tensile failure of the specimen. The splitting tensile strength was determined by dividing the maximum applied load by the appropriate geometrical factors. In terms of a flexural strength test, a one-force load system (3-point bending) was conducted following EN 12390-5 [48] on beam samples with a slot 1/10 h = 1.5 cm high and 3 mm wide. The tests were performed 28 days after curing the beams for PC mixes J1 and J2, J4, J6, J8 standing for plain pervious concrete and modified with polymer fibers, respectively. Each sample was precisely set in the machine with respect to the upper and lower rollers (Figure 5). During the test, a perpendicular point load was applied to the samples at a continuous rate until failure occurred (0.5 mm wide crack). The specimen failed within the middle third of the span length in the tension area or underside of the specimen, and the modulus of rupture was calculated using the following formula:(1)$$f_{b}=\frac{3\times F\times l}{2\times d^{3}},$$
where: f*_b_*—flexural strength (modulus of rupture) [MPa], F—load applied [kN], l—length of the sample (l = 3*d*) [m], *d*—width, height of the sample [m].

Since pervious concrete has a relatively high porosity, it is not suitable to use the submerged weight measurement to obtain its bulk volume. Therefore, a geometrical measurement as an air void test was chosen. The cylindrical samples were cut into 2 cm thick slices to estimate the void ratio (porosity). Each slice surface was scanned and transferred to AutoCAD software (Autodesk Autocad 2022 version S.51.0.0) (see Figure 6). Then, using the image analysis procedure, the area of voids was marked and calculated to achieve total porosity of each sample.

The permeability coefficient test was measured using the falling head method [28]. Permeability coefficient was calculated using Darcy’s law as provided below:(2)$$k=\frac{a\times L}{A\times t}\times \ln\left(\frac{h_{1}}{h_{2}}\right)\;,$$
where: k—coefficient of permeability [cm/s], *a*—cross sectional area of a graduated cylinder [cm^2^], A—cylindrical sample cross section [cm^2^], *L*—length of a sample [cm], *h*_1_—initial water level, *h*_2_—final water level [cm], t—time required for water to fall from level *h*_1_ to *h*_2_ [s]. The test was performed for at least 3 times per sample to obtain an average value of permeability.

According to EN 12390-13 [49], the modulus of elasticity test was carried out in modimeter on cylindrical samples with flat top and bottom surfaces, guaranteeing uniaxial compression. Prior to the test, a series of compressive stress cycles up to approx. 40 percent of the measured compressive strength was applied. The modulus of elasticity of the specimen was corresponded to the average slope of the stress–strain responses captured during cyclic loading. Following ASTM recommendations for the determination of the mean secant modulus of concrete elasticity, the lower stress ($\sigma_{l}$) was taken corresponding to the lower longitudinal deformation ($\varepsilon_{l}=50\times 10^{-6}$) whereas deformation ($\varepsilon_{u}$) corresponded to the upper level of stresses ($\sigma_{u}$) that was equal to 40% of limit compressive strength (0.4 × f_cm_). The modulus of elasticity was figured out using the following formula:(3)$$E_{\mathrm{cm}}=\frac{\sigma_{u}-\sigma_{l}}{\varepsilon_{u}-\varepsilon_{l}}=\frac{\sigma_{u}-\sigma_{l}}{\varepsilon_{u}-0.000050},$$
where: E_cm_—modulus of elasticity [GPa], $\sigma_{u}$—upper level of stresses (0.4 × f_cm_) [MPa], $\sigma_{l}$—lower level of stresses [MPa], $\varepsilon_{u}$—longitudinal deformation corresponding to $\sigma_{u}$, $\varepsilon_{l}$—lower longitudinal deformation ($\varepsilon_{l}=50\times 10^{-6}$).

## 3. Results and Discussion

Herein, the study presented investigated the effects of natural vs. recycled aggregate modified with polymer fibers on characteristics of PC. Eight batches of mixes (J1–J8) were used in the production of pervious concrete (see Table 6).

In general, the unit weight of PC mixes was between 1769 and 2063 kg/m^3^, the void content was between 11.6% and 21.1%, and the permeability was in the range of 0.71–1.24 cm/s (see Figure 7). Since the specimens were made of different aggregate types and quantities, there is a prominent variation in the porosity of the samples and observed consistently high values of permeability, which is dependent on several factors, i.e., tortuosity, porosity, pore size distribution, pore roughness, constrictions of pore space connectivity of internal pore channels, compaction method and times. For any specified application (J1–J8), the obtained value of permeability was high. Therefore, the authors plan, as a next step of the present research, to not only verify the test on the most promising mixtures, but also on specimens prepared with vibrations or the rodding method.

Flexural strength (the-so-called modulus of rupture) is a measure of tensile strength in bending, calculated on Formula (1). Flexural strength is typically used in Portland Cement Concrete mix design for pavements because it best simulates slab flexural stresses, as they are subjected to loading. Because the flexural test involves bending a beam specimen, there is some compression involved, and thus flexural strength shall generally be slightly higher than tensile strength measured using a split tension test. This phenomenon was observed within the natural aggregate component in the PC mixes J1–J2, but no longer noticed in the recycled aggregate composition of mixes J4 and J6 (see Table 6).

The results of the 7, 28 and 56 days compressive strength tests, summarized in Table 6, show that, in all cases, there is a clear increase in strength over time and similar pace when compared to conventional concretes. The general trend is an increase in strength as a function of time, with the highest strength gain in the first seven days. The study yielded a range of values from 17.6 to 21.6 MPa (after 28 days) for PC, which is in line with reported values in the literature. In earlier research with natural aggregate [26], it was found that the compressive strength of PC decreased linearly as the void ratio increased. similar trend was observed in the present research, even though the study was more focused on investigating the impact of recycled aggregate on mechanical properties. Thus, it was found that recycled concrete aggregate (mix J3–J4) slightly influenced the compressive strength and modulus of elasticity, when comparing to natural aggregate concrete (mix J1–J2), which is consistent with previous research [50]—Figure 8. In all tests recycled, concrete aggregate addition increased compressive strength from 4.5 to 17.4% and tensile strength from 2.9 to 21.4%, and from 5.1 to 9.1% for the cubic and cylindrical samples, respectively. This is because recycled aggregate derived from demolished concrete structures, activated by binder, retain proper binding abilities and are able to achieve higher performances [51].

The effect of fibers on pervious concrete behavior was investigated by the incorporation of deliberately used 54 mm in length polymer fibers with a multifilament structure. Mixes J2, J4, J6 and J8 were treated with fibers at constant rate of 2 kg/m^3^. In all mixtures, the addition of fiber produced an increase of 28 days compressive strength from 14.2 to 23.4% (Figure 9). The increment in tensile strength was also noticed, ranging from 8.8 to 25.0%, and from 11.4 to 18.2% for the cubic and cylindrical samples, respectively. The flexural strength test showed an increase in the strength of pervious concrete from 16.0 to 22.5%, with the highest boost recorded in the case of simultaneous use of rubber waste and polymer fibers. The study on the modulus of elasticity showed that the addition of polymer fibers caused an increase of 19.8% and 13.3%, compared to the natural and recycled concrete aggregate composition of PC mixes, respectively (J1 vs. J2 and J3 vs. J4 in Table 6). However, for the above mixes, with the additional use of rubber granules, the module decreased by 4.1% and 11.2%, respectively (J2 vs. J6 and J4 vs. J8 in Table 6). The obtained test results follow the trends set in the past scientific research [22,23,52], apart from the modulus of elasticity. This may be due to the nature, type and size of recycled concrete and rubber waste aggregates used.

The control mixes (J1–J4) had compressive strength in the range of 17.6 to 21.1 MPa, whereas the rubberized mixes (J5-J8) were from 15.6 to 21.6 MPa. In the mixes containing natural coarse aggregate, the compressive strength was decreasing with dosing the rubber waste by a maximum of 11.4% (J1 vs. J5 and J2 vs. J6 in Figure 10). Mixes containing recycled concrete aggregate and rubber waste followed the same trend, however, the decrease of compressive strength was highly counteracted by polymer fibers (J3 vs. J7 and J4 vs. J8 in Figure 10). Present results compared with non-rubberized pervious concrete (control) mixtures are in line with the research [45,53], where it was found that the use of flexible additives significantly aggravated the concrete mechanical properties. Therefore, to compensate for this effect, a binder content shall be increased, or a rubber impregnation shall be applied to improve adhesion to the cement paste.

## 4. Conclusions

The study aimed to evaluate the effects of recycled concrete and rubber waste aggregate addition modified with polymer fibers on characteristics of pervious concrete. Results showed that the use of recycled concrete aggregate (8/31.5 mm) with a replacement ratio of 50% (by weight of aggregate) increased the mechanical properties of pervious concrete in all analyzed cases, whereas the rubber waste aggregate (2/5 mm) with a replacement of 10% (by volume of aggregate) reduced the compressive strength by a maximum of 11.4%. The most satisfactory results were registered by adding 2 kg/m^3^ of polymer fibers, which increased the overall performance of PC, thus enhancing the compressive and tensile strengths by a maximum of 23.4 and 25.0%, respectively. Multifilament structure and a deliberately used length of 54 mm of fibers, proved the strengthening effect of concrete structure by making more effective joints among the large-size grains (max size 31.5 mm). For all PC mixes (J1–J8), the obtained value of permeability was high, ranging from 0.71–1.24 cm/s. The obtained test results are promising and demonstrate the possibility of using the recycled aggregates in the decarbonization process of pervious concrete production.

## 5. Further Directions

In fact, pervious concrete cannot be designed with the only purpose of achieving the greater mechanical performances. To guarantee appropriate drainage and environmental qualities of the material, the optimal balance between strength and permeability is to be targeted. Further laboratory and field performance tests are needed to better understand porosity and its relationship to the mechanistic response variability due to the heterogeneous nature of PC containing recycled aggregate being modified with polymer fibers. Potential research directions may follow, i.e., the clogging potential of pervious concrete, fiber adhesion to the cement paste and influence of dosing and length of fibers, and the possibility of modifying rubber granules to increase adhesion to the cement paste and influence granulate size and its content in the mixture. As mechanical properties of PC are strongly affected by the mixture composition and placement method, further research on different application methods (i.e., compaction, vibration) affecting void content, permeability and strengths is recommended. Therefore, within the next steps of the present study, it is planned to investigate and report on vibrated specimens simulating the potential method of application, with respect to compaction time influencing the strength and porosity of pervious concrete that contains recycled materials.

## Figures and Tables

**Figure 1 materials-16-05222-f001:**
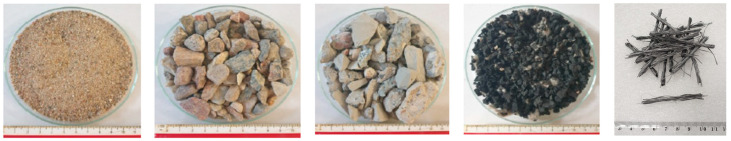
Fine Aggregate (0/2 mm, FA), Natural Coarse Aggregate (8/16 mm, NCA), Recycled Concrete Aggregate (8/31.5 mm, RCA), Rubber Waste Aggregate (2/5 mm, RWA), Polymer Fibers (54 mm length, PF).

**Figure 2 materials-16-05222-f002:**
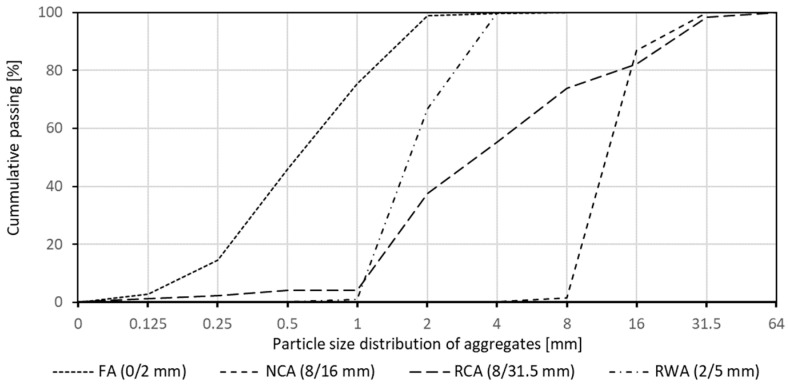
Sieve analysis of aggregates.

**Figure 3 materials-16-05222-f003:**
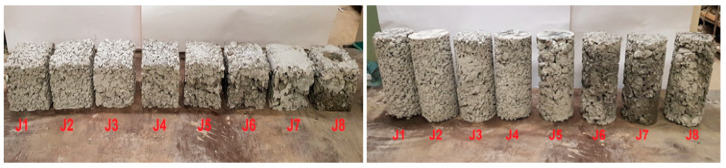
Cubic and cylindrical samples prior to tests—for J1–J8 mixes see Table 4.

**Figure 4 materials-16-05222-f004:**
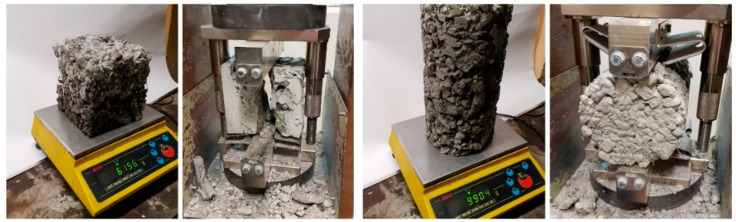
Cubic and cylindrical samples during Tensile splitting strength tests.

**Figure 5 materials-16-05222-f005:**
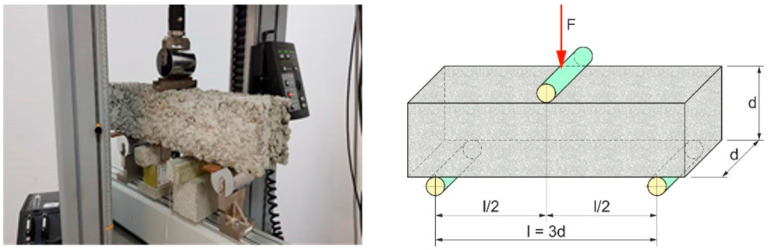
Beam sample during Flexural strength test in a center-point loading flexural testing device INSTRON (Norwood, MA, USA).

**Figure 6 materials-16-05222-f006:**
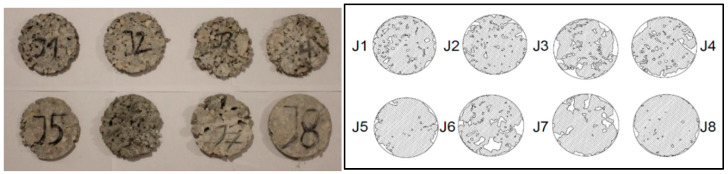
Image analysis procedure of void content (porosity).

**Figure 7 materials-16-05222-f007:**
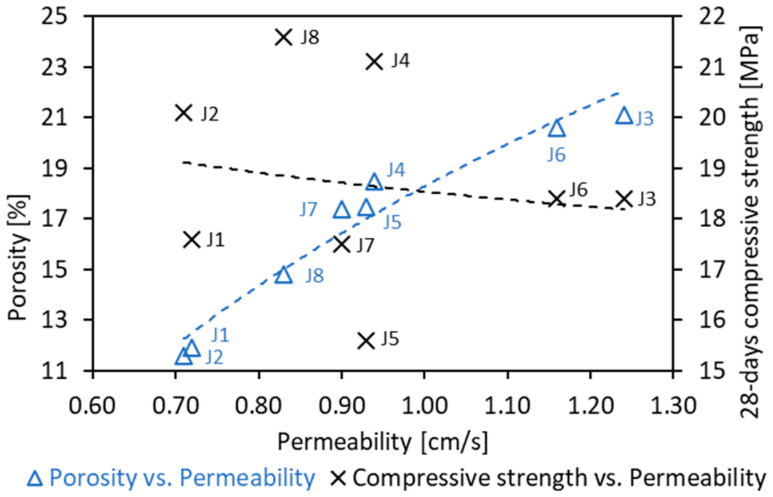
Pervious concrete properties: 28 days compressive strength and void ratio (porosity) vs. permeability (k)—for J1–J8 mixes results see Table 6.

**Figure 8 materials-16-05222-f008:**
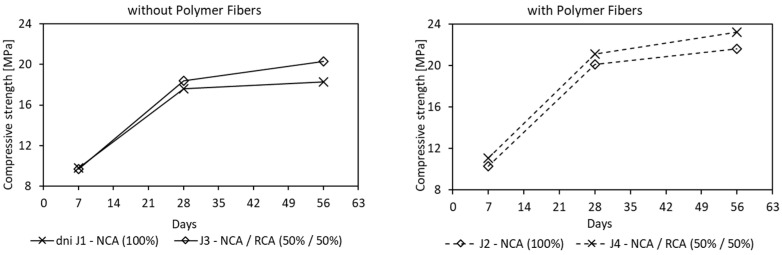
Effect of Recycled Concrete Aggregate on mean compressive strength after 7, 28 and 56 days (Notes: NCA = Natural Coarse Aggregate; RCA = Recycled Concrete Aggregate; RWA = Rubber Waste Aggregate)—for J1–J8 mixes results see Table 6.

**Figure 9 materials-16-05222-f009:**
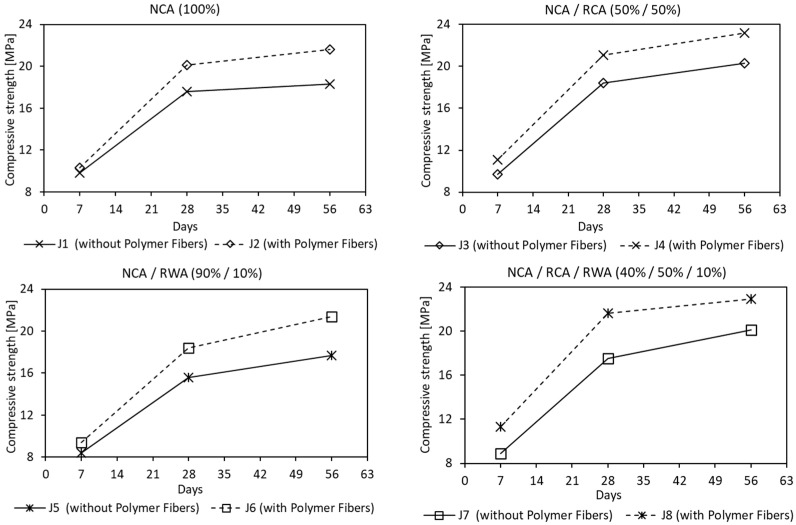
Effect of Polymer Fibers on mean compressive strength after 7, 28 and 56 days (Notes: NCA = Natural Coarse Aggregate; RCA = Recycled Concrete Aggregate; RWA = Rubber Waste Aggregate)—for J1–J8 mixes results see Table 6.

**Figure 10 materials-16-05222-f010:**
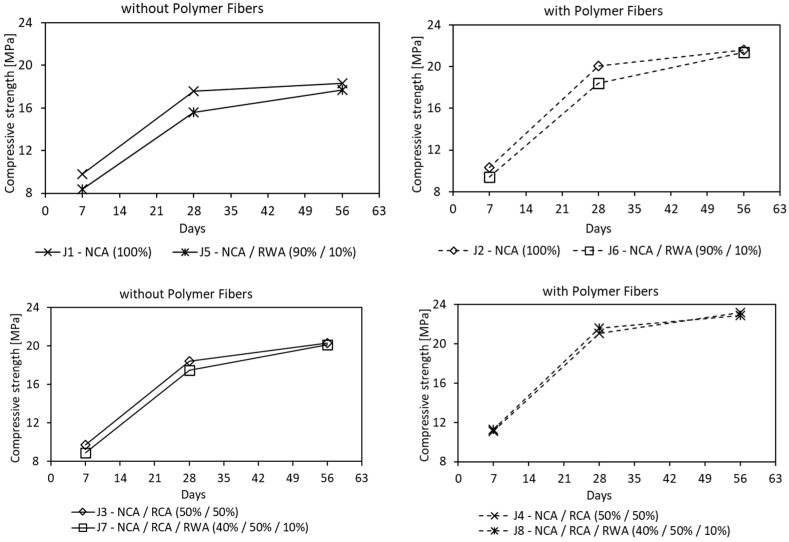
Effect of Rubber Waste Aggregate on mean compressive strength after 7, 28 and 56 days (Notes: NCA = Natural Coarse Aggregate; RCA = Recycled Concrete Aggregate; RWA = Rubber Waste Aggregate)—for J1–J8 mixes results see Table 6.

**Table 1 materials-16-05222-t001:** Properties of pervious concrete.

Void Ratio [%]	Unit Weight [kg/m^3^]	Permeability [cm/s]	28-Day Compressive Strength [Mpa]	Flexural Strength [Mpa]	Tensile Strength [Mpa]	w/b	a/b	Aggregates (Size)	Additive (% by Binder Weight)	Ref.
16.4–20.7	-	0.21–0.26	-	-	-	0.40	4.0–12.0	Coarse	-	[34]
23.0–26.0	1890–1930	0.86–1.02	14.5–17.5	-	1.6–2.0	0.25–0.35	-	Coarse (<8 mm)	-	[35]
26.5–35.1	-	1.5–1.7	-	-	-	0.35	4.5	Gravel (4.75–12.5 mm)	-	[36]
21.0–28.0	-	0.86–2.00	5.1–15.9	1.9–3.2	1.15–1.70	0.35	3.0–7.0	Coarse (4.75–12.5 mm)	-	[37]
-	-	-	13.5	2.8	-	0.41	3.33	Limestone (4.75–12.5 mm)	-	[38]
			19.9	4.4	-				SBR	
			11.5–22.6	4.6–5.8	-				SBR + AP	
-	-	0.35–0.85	18.5–21.5	2.5–3.5	-	0.25–0.35	-	Recycled concrete (5–10 mm)	PPTF 55 mm (0.6–1.5)	[2]
		0.55–0.60				0.30			CCF 12 mm (1.0–1.9)	
		0.45–0.60				0.30			PPF 12 mm (0.1–0.3)	
31.0–34.2	1921–1950	0.18–0.29	9.1–21.5	3.1–4.0	0.54–1.56	0.31	4.8	Coarse	GGBS (0–80)	[39]
-	1800–2500	0.9–1.4	10.0–5.7	-	-	0.35	4.0	Crushed gravel (10 mm)	Fly ash (0–50)	[40]
22.0–28.0	-	-	11.9–15.5	-	-	0.66	10.88	Coarse/Slag (<25 mm)	SBR latex (0.94)	[41]
26.0–28.5			10.0–8.0						NJF (12.5)	
-	2090	0.81	15.1	-	-	0.30	4.4	Coarse	MK (10)	[42]
	2030	1.77	12.2						Fly ash (10)	
	1930	1.05	16.6						UFGGBS (10)	
12.9	2165	-	-	-	-	0.22–0.27	4.1–4.7	River gravel (4.75–9.5 mm)	SBR (10)	[26]
33.0	1790								Silica fume (5)	
-	1769–1929	0.27–1.18	21.4	-	2.40–2.45	0.27	4.7	River gravel (4.75–9.5 mm)	FM300 12.7/19.1 mm (0.10–0.26)	[43]
	1759–1916	0.02–1.03	17.8		1.30–2.05				FC500 50 mm (0.10–0.26)	
-	1778–1985	1.0–1.2	2.6–8.1	-	-	0.30	-	Crushed granite (<20 mm)	EBA (5–25)	[44]
-	1829–1841	0.6–3.0	-	-	0.64–0.69	0.30–0.35	5.0	Coarse	CR 0.6/2.5 mm (5–10)	[17]
	1915–1926				0.34–0.60				FCR 0.08–1 mm (5–10)	
-	1900–2030	0.20–0.33	9.1–14.5	1.14–1.60	-	0.27	2.9–3.3	River coarse (>10 mm)	CTC (10–20)	[45]
	2030–2130	0.15–0.26	13.1–19.0	1.11–1.57					CR 4 mm (10–20)	
	2180–2240	0.26–0.27	14.4–21.6	1.49–1.56					FCR 1 mm (10–20)	

Notes: w/b = water/binder ratio; a/b = coarse aggregate/binder ratio; SBR—Styrene–Butadiene Rubber latex; AP—Acrylate Polymer; PPTF—Polypropylene Thick Fiber; CCF—Copper Coated Steel Fiber; PPF—Polypropylene Fiber; GGBS—Ground Granulated Blast-Furnace Slag; NJF—Natural Jute Fiber; MK—Metakaolin; UFGGBS—Ultra-Fine GGBS; FM300—Fibermesh 300; FC500—Fibercast 500; EBA—Engineered Biomass Aggregate; CR—Crump Rubber; FCR—Fine Crump Rubber; CTC—Coarse Tire Chips.

**Table 2 materials-16-05222-t002:** Sieve size distribution of aggregates.

Type	Sieve Size, % Passing										
	0	0.125	0.25	0.5	1	2	4	8	16	31.5	64
FA (0/2 mm)	0.0	2.9	14.6	45.8	75.5	99.0	99.8	100	100	100	100
NCA (8/16 mm)	0.0	0.1	0.1	0.1	0.1	0.1	0.2	1.6	86.8	100	100
RCA (8/31.5 mm)	0.3	1.1	2.3	4.1	5.2	37.5	55.2	73.8	82.3	98.3	100
RWA (2/5 mm)	0	0.1	0.1	0.1	0.9	66.9	100	100	100	100	100

**Table 3 materials-16-05222-t003:** Chemical composition and physical properties of CEM I 42.5 R.

Initial Setting Time [min]		Final Setting Time [min]		Compressive Strength [MPa]					Blaine Fineness [cm^2^/g]			Loss on Ignition [%]		Water Demand [%]	
				2 Days			28 Days								
155		195		30.2			57.3		3504			3.4		27.5	
Chemical composition [%]															
SiO_2_	Al_2_O_3_		Fe_2_O_3_		CaO	MgO		SO_3_		Na_2_O	K_2_O		TiO_2_		Cl
21.7	6.2		3.1		63.4	1.0		3.9		0.16	0.64		0.25		0.06
Mineral composition [%]															
Na_2_Oeq			C_3_S			C_2_S				C_3_A			C_4_AF		
0.7			63.1			7.6				6.1			8.9		

**Table 4 materials-16-05222-t004:** Material proportions for the laboratory produced pervious concrete mixtures (in kg/m^3^).

	Natural Aggregate		Recycled Aggregate					
Mix ID	J1	J2	J3	J4	J5	J6	J7	J8
Binder (cement)	350	350	350	350	350	350	350	350
Water	120	120	120	120	120	120	120	120
w/b	0.34	0.34	0.34	0.34	0.34	0.34	0.34	0.34
a/b	4.3	4.3	4.3	4.3	3.9	3.9	3.9	3.9
FA (0/2 mm)	90	90	90	90	90	90	90	90
NCA (2/8 mm)	-	-	-	-	-	-	-	-
NCA (8/16 mm)	1520	1520	760	760	1370	1370	610	610
RCA (8/31.5 mm)	-	-	760	760	-	-	760	760
RWA (2/5 mm)	-	-	-	-	50	50	50	50
PF (54 mm)	-	2	-	2	-	2	-	2
Superplasticizer	1.4	1.4	1.4	1.4	1.4	1.4	1.4	1.4

Notes: w/b = water/binder ratio; a/b = coarse aggregate/binder ratio.

**Table 5 materials-16-05222-t005:** Sample types and designed tests.

Sample Type	Quantity	Test
Cube (150 × 150 × 150 mm)	9	Compressive strength Tensile splitting strength
Cylinder (ø150 mm, h = 300 mm)	6	Tensile splitting strength Modulus of elasticity
Beam (700 × 150 × 150 mm)	3	Flexural strength

**Table 6 materials-16-05222-t006:** Test results for the laboratory produced pervious concrete mixtures.

		Natural Aggregate		Recycled Aggregate					
Mix ID		J1	J2	J3	J4	J5	J6	J7	J8
Compressive strength f_cm_ [MPa]	7 days	9.8	10.3	9.7	11.1	8.4	9.4	8.9	11.3
	28 days	17.6	20.1	18.4	21.1	15.6	18.4	17.5	21.6
	56 days	18.3	21.6	20.3	23.2	17.7	21.4	20.1	22.9
Tensile strength [MPa]	cubic	2.9	3.4	3.0	3.5	2.8	3.5	3.4	3.7
	cylindrical	3.2	3.7	3.5	3.9	3.3	3.9	3.6	4.1
Flexural strength [MPa]		3.5	4.1	2.4	2.8	2.5	2.9	4.0	4.9
Unit weight [kg/m^3^]		2034	2063	2024	1970	1779	1769	1975	1822
Modulus of elasticity E_cm_ [GPa]		10.6	12.7	10.9	12.4	11.2	12.2	8.6	11.0
Void ratio (porosity) [%]		11.9	11.6	21.1	18.5	17.5	20.6	17.4	14.8
Permeability, k [cm/s]		0.72	0.71	1.24	0.94	0.93	1.16	0.90	0.83

## Data Availability

The datasets generated, used and/or analyzed during the current study are included in this published article. All raw data available from the corresponding author on reasonable request.

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
