# Peer review of "Influence of the Addition of Recycled Aggregates and Polymer Fibers on the Properties of Pervious Concrete"

_materials, 2023, doi:10.3390/ma16155222_

Round 1

Reviewer 1 Report

The article presents a study on the incorporation of recycled rubber and fibers in permeable concrete. The incorporation of recycled rubber in concrete is not a new subject, but new results were presented and should be discussed. Some notes are presented:

1# Permeable concrete is a concrete that does not contain cement in its fine aggregate mixture. The authors work with an abundant amount of fine sand, which will certainly generate a concrete with low permeability. I suggest that the authors review the concrete mix, as it is not easy to believe that this mix gave this result;

2# In chapter 2, the authors present the materials used. The fine sand used by the authors states that it is 0 mm in diameter. Authors must include a grading curve for all materials, including recycled rubber;

3# In the summary, the authors report that the mixture with 50% rubber had improved concrete performance. But in the results this does not match. Stamina resistance has been lowered, as is obvious. Authors should revise and be more assertive in their statement in the abstract;

4# In conclusion, the authors state that mixtures with fibers had an increase in resistance to resistance, but they did not based on the results. Authors must review the entire conclusion.

Authors must make these corrections in the article.

Author Response

Reviewer 1

Thank you for all the comments and suggestions that contributed to the improvement of the article and helped us in a deeper analysis of the issues that we tried to present in our work.

Hereby, we refer to Your comments:

1# Permeable concrete is a concrete that does not contain cement in its fine aggregate mixture. The authors work with an abundant amount of fine sand, which will certainly generate a concrete with low permeability. I suggest that the authors review the concrete mix, as it is not easy to believe that this mix gave this result;

Ad. 1#

Properly designed pervious concrete should enable a significant flow rate of water through the pervious concrete mass. Pervious concrete the-so-called No-Fines Concrete has little or no fine aggregate in the mixture. No fine content is often used in the case of crushed aggregate. However, in case of natural pebble aggregate to increase mechanical properties, fine aggregate is used in small amounts 80-140 kg/m3. Analysis of numerous studies results indicates that many researchers have used this solution (Table 1). Other studies have shown that if natural aggregate with a grain size of 16-31.5mm was used (without taking into account cement grout and sand), the volume of voids was 43-45%, while for the crushed aggregate with the same fraction (16-31.5mm), the void ratio was lower and ranged from 33-37%. Therefore, if we use 90kg/m3 of sand (0-2mm) and 50kg/m3 of granulated rubber, the total volume of these components will not exceed 56 dm3, which results in <1% of the total volume of the mixture together with the pores. Summing up, the use of pebble aggregate and fine aggregate or other small-size additives in the total volume of up to 5% does not result in a significant reduction in permeability but may significantly improve the mechanical properties of pervious concrete (Wang et al. 2006, Onstenk et al. 1993, ACI Committee 522 Report 2010).

2# In chapter 2, the authors present the materials used. The fine sand used by the authors states that it is 0 mm in diameter. Authors must include a grading curve for all materials, including recycled rubber;

Ad. 2#

Thank You for the comment. We added a sieve analysis curve for all materials, including recycled rubber (see Figure 2, lines 133-134).                            

3# In the summary, the authors report that the mixture with 50% rubber had improved concrete performance. But in the results this does not match. Stamina resistance has been lowered, as is obvious. Authors should revise and be more assertive in their statement in the abstract;

Ad 3#

Thank You for the comment. The authors investigated various effects of pervious concrete properties changes due to incorporation of Recycled Concrete Aggregate (RCA) or Rubber Waste Aggregate (RWA) as a partial replacement for Natural Coarse Aggregates (NCA). Results in chapter 4 (Conclusions) are summarized for RCA replacement (lines 303-306) and RWA replacement (lines 306-308), respectively.

4# In conclusion, the authors state that mixtures with fibers had an increase in resistance to resistance, but they did not based on the results. Authors must review the entire conclusion. Authors must make these corrections in the article.

Ad 4#

Thank You for the comment. The authors investigated the effect of adding polymer fibres to mixtures J2, J4, J6 and J8 according to Table 4. Their effect on pervious concrete mixtures is presented in detail in Figure 9 and in the discussion of the obtained results (lines 270-285) and in Chapter 4 (Conclusions) – lines 314-316. The conclusions section has been accordingly modified.

Reviewer 2 Report

The article deals with a very interesting topic related to the effects of recycled concrete and rubber waste aggregate modified with polymer fibers of pervious concrete. I have some comments and questions to help improve the article to make it more complete:
1.     What is the motivation fixed w/b 0.34?
2.     It is confusing to the readers what is authors want to say in the abstract and conclusion (i.e., recycled concrete aggregate (8/31.5 mm), rubber waste aggregate (2/5 mm)).
3.     Correct the unit “kg/m3” in line 79.
4.     At the last paragraph of the Introduction Section, the authors should present more clear research gap and novelty of this work.
5.     Correct the sentence in line 119.
6.     It is observed that the w/b ratio for the control concrete is 0.5, whereas the pervious concrete is 0.34. Justify test results comparison of control and pervious concrete is accurate.
7.     How do you measure void ratio using the AutoCAD image provide a reference in Section 2.4, page 4?
8.     Write the meaningful exponential function (not like εl = 50 ∙ 10-6) in lines 217 and 222.
9.     In Figure 7, use black font for axis descriptions and provide legends. It wonders about the axis value (i.e., 0,50; 0,75; 0,100).
10.  In Figure 8, why did not compare results with the control sample?
11.  Write the axis descriptions of compressive strength in Figures 9, 10, and 11.
12.  In Figures 9 and 11, How did you select the pair of different mixes to justify it?

13.  The conclusion section is lack of logic. Please logically present the main findings.

Minor editing of English language required. Some spelling mistakes need to be corrected carefully. 

Author Response

Reviewer 2

Thank you for all the comments and suggestions that contributed to the improvement of the article and helped us in a deeper analysis of the issues that we tried to present in our work.

Hereby, we refer to your comments:

  1. What is the motivation fixed w/b 0.34?

Ad.1.

Initial mixes and previous research were analyzed prior to the experiment to evaluate the feasible mixing ratios. Water to binder ratio was fixed then at 0.34 (see lines 111-113).

  1. It is confusing to the readers what is authors want to say in the abstract and conclusion (i.e., recycled concrete aggregate (8/31.5 mm), rubber waste aggregate (2/5 mm)).

Ad.2.

Thank You for the comment. The title, abstract and conclusions sections were accordingly modified.

  1. Correct the unit “kg/m3” in line 79.

Ad.3

Thank You for the comment. Corrected (see line 81).

  1. At the last paragraph of the Introduction Section, the authors should present more clear research gap and novelty of this work.

Ad.4.

Thank You for the comment. The introduction section has been modified (see lines 101-116).

  1. Correct the sentence in line 119.

Ad.5

Thank You for the comment. Corrected (see lines 126-128).

  1. It is observed that the w/b ratio for the control concrete is 0.5, whereas the pervious concrete is 0.34. Justify test results comparison of control and pervious concrete is accurate.

Ad.6

Thank You for the comment. The idea of testing control concrete (C1-C3 mixes) was to check overall performance of the mixture with additives ie. Rubber waste, polymer fibers for specific applications in geotechnics. In fact it shouldn’t and hasn’t been compared nor correlated to any pervious concrete results, thus all data concerning control concrete is erased from the publication.  

  1. How do you measure void ratio using the AutoCAD image provide a reference in Section 2.4, page 4?

Ad.7

Thank You for the comment. The cylindrical samples were cut into 2-cm thick slices to estimate the void ratio (porosity). Each slice surface was scanned and transferred to AutoCAD software. The authors used manual options of the cad software, no automatic procedure was available. Then, the regions with enclosed areas were drawn and calculated. Based on these measurements, the porosity of each sample was determined.

As authors, we are aware that this is only an estimation measurement, that is why in the next stage of research we intend to use other methods of measuring porosity i.e. X-RAY MICRO CT.

  1. Write the meaningful exponential function (not like εl = 50 ∙ 10-6) in lines 217 and 222.

Ad.8

Thank You for the comment. Corrected (see line 216 and 221).

  1. In Figure 7, use black font for axis descriptions and provide legends. It wonders about the axis value (i.e., 0,50; 0,75; 0,100).

Ad.9

Thank You for the comment. Figure 7 has been modified.

  1. In Figure 8, why did not compare results with the control sample?

Ad.10

Thank You for the comment. Figure 8 reproduced the data from table 6 in a graphical form and is not necessary in the publication, thus has been erased.

  1. Write the axis descriptions of compressive strength in Figures 9, 10, and 11.

Ad.11

Thank You for the comment. Figure 9, 10 and 11 has been modified.

  1. In Figures 9 and 11, How did you select the pair of different mixes to justify it?

Ad.12

Figure 9 examined the effect of adding recycled concrete (J3, J4, J7 and J8) to the corresponding mixtures on natural aggregate. In Fig. 11, the effect of adding rubber granules (J5-J8) to the corresponding mixtures without the addition of rubber was checked. The upper two fields of the chart were missing a legend – it has been corrected. In addition, the graphs were divided into those mixtures that contained and did not contain polymer fibers.

  1. The conclusion section is lack of logic. Please logically present the main findings.

Ad.13

Thank You for the comment. The conclusions section and title has been accordingly modified.

Comments on the Quality of English Language

  1. Minor editing of English language required. Some spelling mistakes need to be corrected carefully. 

Ad.14

Thank You for the comment. Spelling has been carefully checked and corrected.

Reviewer 3 Report

The article provides a good literature review on the problem raised; a large amount of research has been carried out; interesting results have been obtained

Author Response

Reviewer 3

Thank you for all the comments and suggestions that contributed to the improvement of the article and helped us in a deeper analysis of the issues that we tried to present in our work.

Hereby, we refer to your comments:

  1. It is necessary to add information about the relevance of the research presented in the article.

It is recommended to specify more clearly that the main object of research is not the improvement of

the properties of permeable concrete, but the possibility of recycling waste (secondary aggregate and

crumb rubber) in the manufacture of this concrete.

Ad.1

Thank You for the comment. The abstract, title and conclusions section has been accordingly modified.

  1. Object research would be better moved from the Methods and Materials section to the Introduction

Ad.2

Thank You for the comment. Section “Research objectives and scope” has been modified and moved to the Introduction chapter (see lines 101-116).

  1. (L113). Based on what data was the amount of mixing water fixed at 0.34?

Ad.3

Thank You for the comment. Initial mixes and previous research were analyzed prior to the experiment to evaluate the feasible mixing ratios. Water to binder ratio was fixed then at 0.34 (lines 111-113).

  1. In Figures 8, 9, 10, 11, you should indicate the percentage deviations of the values (deviation bars).

Ad.4

Thank You for the comment. The data in Fig.9-11 has been updated. Due to other reviewer’s comment - Figure 8 reproduced the data from table 6 in a graphical form and is not necessary in the manuscript, thus has been erased.

  1. Table 2. Why do the authors provide information on the results of sifting fibers that have not yet been

divided into individual fibers? It would be better to remove this information.

Ad.5

Thank You for the comment. Table 2 represents sieve analysis for Fine Aggregate (0-2 mm, FA), Natural Coarse Aggregate (8-16 mm, NCA), Recycled Concrete Aggregate (8-31.5 mm, RCA), Rubber Waste Aggregate (2-5 mm, RWA). A sieve analysis graph has been added (see Figure 2)

  1. (L 130). Specify which superplasticizer was used in the research. Please explain whether the use of

a superplasticizer does not lead to a significant decrease in the yield strength of the cement

paste (adhesive) and its runoff from the aggregate particles?

Ad.6

Thank You for the comment. The type of superplasticizer has been included in lines 138-139. We agree with the comment. Preliminary studies have shown that the use of plasticizer in the optimal amount improves the adhesion of the grout to fibers, rubber granules and aggregate. Too much plasticizer would cause the grout to run off, so the determination of the content of the admixture was preceded by previous attempts.

  1. (L 133). The authors consider polymer fiber as a possible alternative to steel, but do not indicate what material it is made of and what its modulus of elasticity is. Many types of polymer fibers have a low modulus of elasticity and are not able to increase the tensile strength of concrete.

Ad.7

Thank You for the comment. In lines 143-146 a specific type of fibers has been included with its parameters. However, it should be noted that we are aware that steel fibers have different properties, which is why our main goal was to use corrosion-resistant polymer fibers as an alternative. In our opinion, fibers of this type are suitable for permeable concrete.

  1. Figure 6. All Figures are shown with the same visible field size, so it is not clear why the appearance of

the fiber in Figures 2 and 3 is so different? At what point does the fiber fragments separate into individual fibers?

Ad.8

Thank You for the comment. Multifilament fibers separate at the time of mixing with coarse aggregate into fine fibers. This is their positive feature, which allows for an even distribution of fibers in the concrete structure, including pervious concrete. However, due to other reviewer’s comment - Figure 6 has been erased from the manuscript.

  1. (L 289). The permeability of concrete during operation largely depends not only on the initial porosity, but on the gradual clogging of the channels for the passage of water. What do the authors think about the effect of fibers on this process, given that the fibers can be stretched across the channel and trap debris?

Ad.9

Thank You for the comment. Several research has been made on defining clogging potential for pervious concrete affecting permeability. The authors plan to further develop a mixture of pervious concrete with the addition of recycled aggregates modified with polymer fibers and further research, including recognizing the clogging effect.

Reviewer 4 Report

This article focuses on previous concrete performance. This manuscript is well organized. However, some major issues need to be addressed before publication.

Abstract: Authors can condense the abstract and include major outcomes (phases of development in the blends) of the study

Line 79: read as kg/m3

Table 1: 2nd column heading … read as Unit weight (kg/m3)

Section 2.1: move this section to the end of the introduction. Please highlight the research significance.

Please mention the properties of rubber aggregates and compare them with natural aggregates.

Line 128: Mention details about the superplasticizer

Figure 8: Authors should present error bars.

Line 301: strengthen the statement with previous literature by highlighting the role of flexible material in the concrete matrix https://doi.org/10.1016/j.cscm.2023.e02200.

Need more explanation of concrete matrix?

The conclusion should be rewritten and made briefer to show the major findings of the study and the conclusions drawn.

Minor editing of English language required.

Author Response

Reviewer 4

Thank you for all the comments and suggestions that contributed to the improvement of the article and helped us in a deeper analysis of the issues that we tried to present in our work.

Hereby, we refer to your comments:

  1. Abstract: Authors can condense the abstract and include major outcomes (phases of development in the blends) of the study

Ad.1

Thank You for the comment. The abstract has been accordingly modified.

  1. Line 79: read as kg/m3

Ad.3

Thank You for the comment. Corrected (see line 81).

  1. Table 1: 2nd column heading … read as Unit weight (kg/m3)

Ad.3

Thank You for the comment. Corrected.

  1. Section 2.1: move this section to the end of the introduction. Please highlight the research significance.

Ad.4

Thank You for the comment. Section “Research objectives and scope” has been modified and moved to the Introduction chapter (see lines 101-116).

  1. Please mention the properties of rubber aggregates and compare them with natural aggregates.

Ad.5

Thank You for the comment. A sieve analysis graph has been added (see Figure 2)

  1. Line 128: Mention details about the superplasticizer

Ad.6

Thank You for the comment. The type of superplasticizer has been included in lines 138-139.

  1. Figure 8: Authors should present error bars.

Ad.7

Thank You for the comment. Due to other reviewer’s comment - Figure 8 reproduced the data from table 6 in a graphical form and is not necessary in the manuscript, thus has been erased.

  1. Line 301: strengthen the statement with previous literature by highlighting the role of flexible material in the concrete matrix https://doi.org/10.1016/j.cscm.2023.e02200.

Ad.8

Thank You for the comment. The reference has been added to the manuscript (see line 303).

  1. Need more explanation of concrete matrix?

Ad.9

Thank You for the comment. The term "cement matrix" has been changed to "cement paste".

  1. The conclusion should be rewritten and made briefer to show the major findings of the study and the conclusions drawn.

Ad.10

Thank You for the comment. The conclusion section has been accordingly modified.
